# Prevalence and factors associated with *Trichomonas vaginalis* infection in indigenous Brazilian women

**Marcelo dos Santos Barbosa**[1], **Iara Beatriz Andrade de Souza**[1], **Erica Cristina dos Santos Schnaufer**[1], **Liliane Ferreira da Silva**[2], **Crhistinne Carvalho Maymone Gonçalves**[3], **Simone Simionatto**[1], **Silvana Beutinger Marchioro**[1,4]*

1 Laboratório de Pesquisa em Ciências da Saúde, Universidade Federal da Grande Dourados—UFGD, Dourados, MS, Brazil, 2 Distrito Sanitário Especial Indígena do Mato Grosso do Sul, Campo Grande, MS, Brazil, 3 Secretária Estadual de Saúde do Mato Grosso do Sul, Campo Grande, MS, Brazil, 4 Laboratório de Imunologia e Biologia Molecular, Instituto de Ciências da Saúde, Universidade Federal da Bahia, Salvador, BA, Brazil

* silmarchioro@hotmail.com

**Data Availability Statement:** All relevant data are within the paper and its Supporting Information files.

## Abstract

There is a scarcity of studies on the prevalence of *Trichomonas vaginalis* (TV) in indigenous populations of Brazil. We conducted a cross-sectional study between January and December 2018, on indigenous women living nearby an urban center of the Midwest region of Brazil and determined the prevalence of TV. Factors associated with TV infection and a comparison of molecular and direct microscopy diagnoses were determined. 241 indigenous women aged above 18 years participated in the study. Cervical and vaginal brush samples were collected to diagnose TV through polymerase chain reaction (PCR). Direct microscopy for detection of TV, and cellular changes was performed. A sociodemographic and behavioral questionnaire was applied at the beginning of the study. All the data were analyzed using Statistical Package for the Social Sciences. The result obtained showed that 27.8% [95% CI: 22.2–33.9] were positive for TV on PCR, while 7.41% [95% CI: 4.1–11] showed positive on direct microscopy. Direct microcopy also found 21 (8.71%) and 8 (3.31%) women infected with *Gardnerella vaginalis* and *Candida albicans*, respectively. In addition, 10 women presented atypical squamous cells of unknown significance and 14 lesions suggestive of HPV. Single women, under the age of 30 and who do not use condoms, were found to have a greater chance of getting TV infection. The high prevalence TV found in this population is comparable to highly vulnerable populations, as prisoners, sex workers and women in regions with low socioeconomic levels, moreover, seems to be an underdiagnosis of this infection. Therefore, a routine test program, as well as a review of the diagnostic method used, is encouraged for proper management.

**Funding:** This work was partially supported by the National Council for Scientific and Technological Development (CNPq grant 440245/2018-4), Support Foundation for the Development of Education, Science and Technology of the State of Mato Grosso do Sul (FUNDECT grants 092/2015 and 041/2017), Coordenação de Aperfeiçoamento de Pessoal de Nível Superior- Brazil (CAPES, Finance code 001), Government of the State of Mato Grosso do Sul, Health Department of the state of Mato Grosso do Sul and Proex/ Universidade Federal da Grande Dourados (UFGD). The sponsors had no role in the collection, analysis, and interpretation of data or the writing of the manuscript.

**Competing interests:** The authors have declared that no competing interests exist.

## Introduction

Developing countries are mainly threatened by infectious diseases of the reproductive tract and sexually transmitted infections (STIs), which are considered a major public health problem. The main and most prevalent STIs described in the literature are caused by *Chlamydia trachomatis*, *Neisseria gonorrhoeae*, *Trichomonas vaginalis* (TV), and *Treponema pallidum* [1, 2]. These pathogens can cause acute urogenital conditions, such as cervicitis, urethritis, vaginitis, and genital ulceration [1, 2]. Trichomoniasis, caused by TV, is the non-viral STI most common in the world [3, 4]. It is a flagellated protozoan affecting both sexes, and women are more affected comparatively. Approximately 85% of women are asymptomatic carriers of TV. However, once the symptoms arise, it affects the region of the vulva, vagina, and cervix, being associated with pelvic inflammation, neoplasms, premature birth, and the promotion of HIV infection [4].

The diagnosis of TV in the public health network is basically performed through direct microscopy of a sample obtained in the Pap smear test. Although the Pap smear has specificity for *T. vaginalis*, there are sensitivity limitations that prevent the use of this technique for the diagnosis of infection [3, 5]. The main focus of this technique is on the search for cellular changes such as atypical granular cells and squamous lesions, other findings such as *Gardnerella vaginallis* infection or *Candida albicans*, end up being in the background, which can also be related with problems in the female reproductive tract, especially the imbalance of bacterial flora, increasing the risk of infection by other STIs [6–8].

In countries like Brazil, trichomoniasis remains unreported, making it difficult to measure the extent of the disease [9]. In 2012, the World Health Organization (WHO) reported about 357 million new cases of non-viral STIs worldwide, of which 142 million were caused by TV [4]. Studies conducted in Brazil have reported TV infection prevalence rates between 3.7% and 30% [4, 10]. Many populations related to low socioeconomic conditions, along with other factors such as age, gender, and sexual choice, have shown above-average prevalence rates for STIs [4, 11]. *T. vaginalis* mainly affects the poorest sections of the population and people in vulnerable situations [12].

An estimated indigenous population of 817 thousand people from 305 different ethnicities lives in Brazil. This population is divided into 5,614 villages, occupying approximately 12.6% of the Brazilian territory, most of them living in lower socioeconomic conditions than the rest of the country [13–15]. Although Brazil has specific legislation for indigenous health, only little is applied. Research on this population is still primarily related to sociocultural issues, and very little has been published on health issues [14]. In this context, the prevalence of the TV was determined in indigenous women living in the Midwest region of Brazil. We determined the factors associated with TV infectious as well as performed a comparison of molecular and direct microscopy diagnoses used in primary care.

## Methods

### Type of study and sample size calculation

A cross-sectional study was conducted to determine the prevalence of TV in indigenous women from two Indigenous Reserves of Dourados—Mato Grosso do Sul, in the Midwest region of Brazil. We obtained approval from the National Research Ethics Commission (number 2.000.496) for this study. Based on the population of 8,000 women aged above 18 years, considering an average prevalence of *T. vaginallis* of 15% (4–27%) and a 5% margin of error, a minimal sample of 192 women was estimated [12].

## Study participants and epidemiological questionnaire

The study was performed between January 2018 and December 2018. A total of 241 women, over the age of 18, agreed to participate in the study and signed an informed consent form. Data and samples were obtained during their visit to the Basic Health Units of the two indigenous reserves evaluated: Bororó and Jaguapiru village, where Guarani (Kaiowá and Nhandeva) and Terena ethnicities predominate.

A questionnaire addressing demographic was given to the participants contemplating the following variables: village (Bororó or Jaguapirú), and ethnicities (Guarani-Kaiowa, Guarani Nhandeva, Terena, and others not predominant in the region); socioeconomic data (age, employment status, income, government benefits, and years of schooling); clinical data: history of cancer, symptoms, and other STIs diagnoses; and behavioral factors: marital status, number of partners in the last year, alcohol use, tobacco use, illicit drug use, share objects (syringes and personal hygiene supplies), condom use. Categorical variables were represented as "yes" or "no", and numerical variables were categorized later.

All possible actions were taken to reduce bias during data collection. For easy comparisons and identification of errors and inconsistencies, questionnaires were filled out in Redcap with double data entry. Questionnaires were redone to resolve inconsistencies. Incomplete questionnaires were discarded before laboratory analysis.

The exams results were submitted to the Health Unit, where the necessary treatment measures could be taken for the patients in whom an infection was detected.

## Sampling and DNA extraction

Vaginal samples were collected using a VAGISPEC® cervical and vaginal brush and placed in a tube containing 1 mL of extraction buffer (EB; 10 mM Tris-HCl pH 8.5, 1 mM EDTA). A total of 300 µL of samples was added to 500 µL of lysis buffer (15 mM sodium citrate, 450 mM NaCl, 0.2% SDS, and 2 mg/mL lysozyme). Phenol-chloroform was used for DNA extraction, and 95% ethanol and 100 µL of 3 M sodium acetate (pH 5.2) were used for precipitation. The precipitate was oven-dried, eluted in 30 µL of TE buffer (10 mM Tris HCl [pH 7.6], 1 mM EDTA), and then frozen at –80˚C. DNA was quantified using the UV-Vis Biodrop µLite Spectrophotometer (Biochrom, USA). DNA integrity was evaluated by agarose gel and by amplification of β-globin gene (268 base pairs) representing the internal control of the reaction.

## Molecular diagnosis, and direct microscopy of *T. vaginalis*

All procedures for polymerase chain reaction (PCR), including primers, were performed according to the one described by Gatti *et al.* (2017). The primer set TVK3/7 (5′ AT TGT CGA ACA TTG GTC TTA CCC TC-3′/5`-TCT GTC CCG TCT TCA AGT ATG C-3`), which amplifies a 260 bp DNA fragment, along with a commercial kit: Master Mix (LUDWIG)® was used. The samples were resolved by 1.5% agarose gel electrophoresis and detected by ethidium bromide staining under ultraviolet light for the confirmation of the results. For all PCR assays, *T. vaginalis* DNA extracted from the culture of ATCC 30236 strain was used as a positive control. Sterile water was used as a negative control to monitor crossover contaminations. The DNA extracted from a PCR-negative sample was used as a negative control [4]. PCR-positive samples were sequenced to confirm the molecular diagnosis in the CREBIO-Jaboticabal/SP (Centro de Biotecnologia, Jaboticabal, SP, Brazil). These sequences were analyzed using Bio Edit software and aligned on BLAST to obtain sequence homology with *T. vaginalis*.

The samples were smeared directly on the slide using a cervical and vaginal brush. The slides were heat-fixed, Giemsa stained, and then examined under 10x and 40x magnification in a microscope. Trained and experienced personnel processed the readings of the slides. The

microscopy also evaluated, besides TV infection, cellular changes, that may indicate HPV infection, as well as the presence of other microorganisms such as *C. albicans* and *G. vaginalis.*

## Statistical analysis

All the data were analyzed using Statistical Package for the Social Sciences (SPSS) version 22.0 (IBM, Armonk, USA). Descriptive statistics were performed to examine the categorical variables and the results presented in proportions (%). A chi-squared test was used to compare categorical variables. The $p < 0.05$ was considered statistically significant. A one-tailed Z test was done to test the differences in proportions. The Clopper–Pearson exact test was done to calculate the binomial confidence interval. The independent contribution of each variable toward the chance of trichomoniasis being present was determined by multivariate analysis. All variables of the bivariate analysis were tested to construct the model. After controlling for all potential confounders, binary logistic regression was used to evaluate the association between TV infection and sociodemographic factors at $p < 0.05$, and they were presented as odds ratio (ORs) together with confidence intervals (95% CI).

## Results

### Sample profile and prevalence of TV

Of 283 samples, 42 (23: questionnaire failure; 19: sampling failure) were excluded. Therefore, 241 indigenous women were screened for the study (Fig 1). The baseline characteristics showed that 81 (33.6%) were aged 18 to 29 years, 200 (83%) were less educated (less than eight years in school), 173 (71.78%) belonged to a family with income below USD 189.00 (which corresponds to minimum Brazilian wage), and 167 (69.29%) were dependent on government benefits (Family Grant). One-hundred and sixty-six (68.88%) women were from the Bororó village; of these, 73.86% were from the Guarani-Kaiowá ethnic group.

An overall prevalence of TV infection using the primer set TVK3/7 was 27.8% (67/241) [95% CI: 22.2–33.9], whereas this rate dropped to 7.41% (17/241) [95% CI: 4.1–11.0] on direct microscopy (Table 1).

### Direct microscopy findings

Direct microscopic analysis showed the presence of *G. vaginalis* and *C. albicans* in 21 (8.71%) [95% CI: 5.5–13.0] and 8 (3.31%) [95% CI: 1.4–6.4] women, respectively. In addition, human papillomavirus (HPV) infected lesions and atypical squamous cells of unknown significance (ASC-US) were observed in 14 (5.81%) [95% CI: 3.21–9.55]) and 10 (4.14%) [95% CI: 2–7.5]) women evaluated. The presence of concomitant infections was observed in 6 of 67 women, 3 (4.47%) with TV and *G. vaginalis*; 3 (4.47%) with TV and *C. albicans*, and 5 (7.45%) with TV and lesions suggestive of infection by HPV and ASC-US (Table 1).

### Sociodemographic status and behavior characteristics

The highest proportions of TV-positive women belonged to age groups 18 to 29 years 35.08% [95% CI: 26.22–46.67], from Bororo village 33.13% [95% CI: 26.03–40.85], 34.78% [95% CI: 21.35–50.25] were unemployed, and 31.53% [95% CI: 24.76–39.37] depended on government benefits. A high prevalence of TV infection was seen in single women 39.29% [95% CI: 27.58–52.37], 34.21% [95% CI: 21.21–50.11] of them had multiple sex partners; 29.3% [95% CI: 23.31–35.88] did not use condoms, and 60% [95% CI: 14.66–94.73] had previous history of STIs. In addition, the two women who used illegal drugs were TV positive (Table 2).

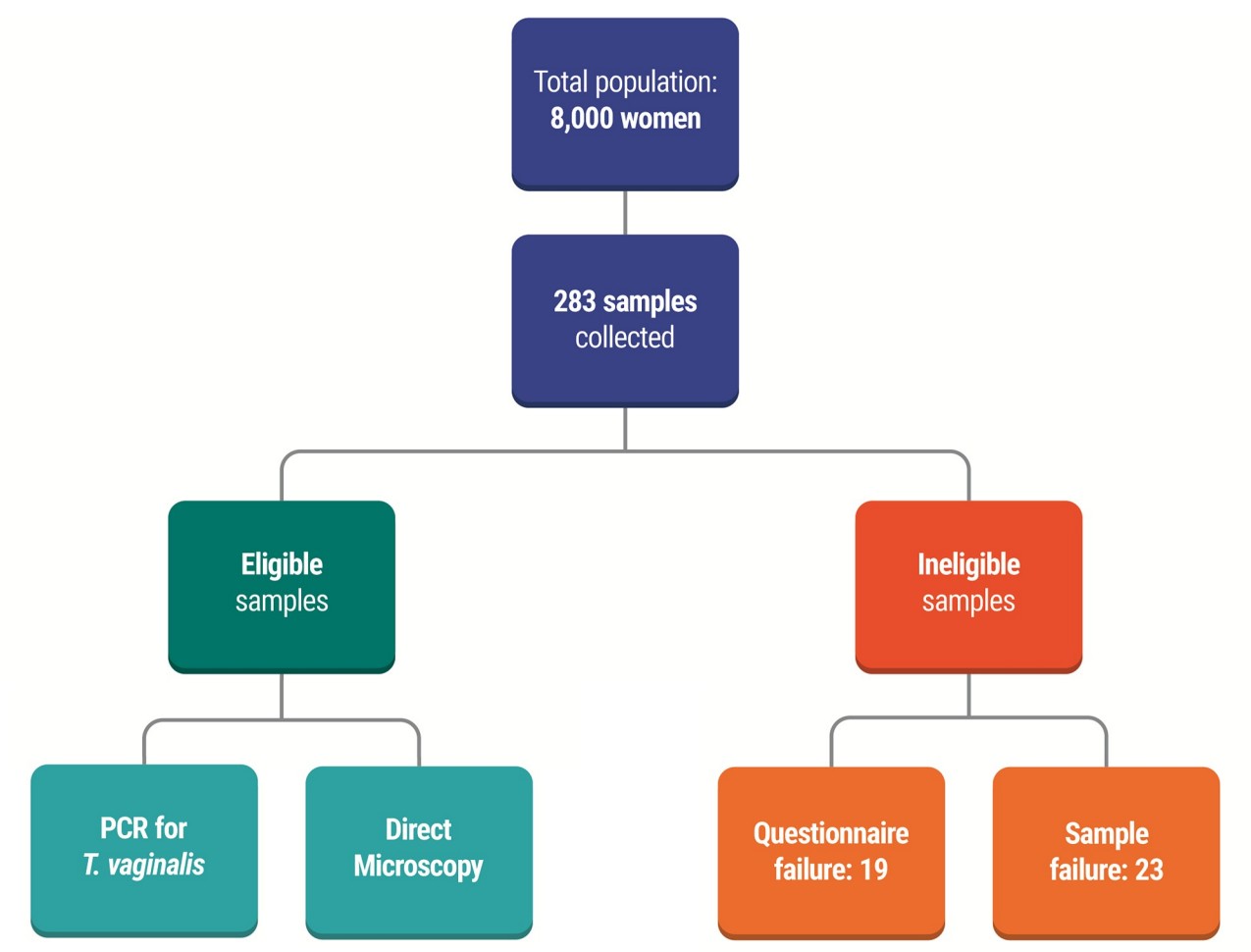

**Fig 1. Sampling flowchart showing the number of samples collected and diagnosis methods performed.**

## Factors associated with TV infection

Additional analysis of the possible risk factors for TV infection showed that the indigenous women who live in the Bororó village have almost three times more risk of infection

**Table 1. Prevalence of *T. vaginalis* by PCR and direct microscopy and others cytologic finds in direct microscopy in indigenous women from Dourados/MS reserve.**

|  | n | % | 95% CI* |
|---|---|---|---|
| *T. vaginalis* (PCR) | 67 | 27.80 | 22.20–33.90 |
| *T. vaginalis* (DM) | 17 | 7.41 | 4.10–11.00 |
| *G. vaginalis* (DM) | 21 | 8.71 | 5.50–13.00 |
| *C. albicans* (DM) | 8 | 3.31 | 1.40–6.40 |
| **Cytologic change** | **24** | **9.96** | **6.49–14.45** |
| ASC-US (DM) | 10 | 4.14 | 2.00–7.50 |
| Suggestive HPV (DM) | 14 | 5.81 | 3.21–9.55 |

*Clopper-Pearson test. PCR: Polymerase Chain Reaction. DM: Direct Microscopy, ASC-US: atypical squamous cells of unknown significance.

**Table 2. Prevalence of *T. vaginalis* in indigenous women in relation to sociodemographic and behavioral variables.**

| Variable | | N | % | Positive | Negative | TV % | 95% CI* |
|---|---|---|---|---|---|---|---|
| **Socioeconomic characteristics** | | | | | | | |
| Age (years) | | | | | | | |
| | 18–29 | 81 | 33.60 | 29 | 52 | 35.80 | 26.22–46.67 |
| | 30–39 | 66 | 27.38 | 19 | 47 | 28.79 | 18.30–41.25 |
| | 40–49 | 50 | 20.74 | 14 | 36 | 28.00 | 16.23–42.49 |
| | 50–59 | 33 | 13.69 | 5 | 28 | 15.15 | 5.11–31.90 |
| | ≥60 | 11 | 4.56 | 1 | 10 | 9.09 | 0.23–41.28 |
| Village | | | | | | | |
| | Bororó | 166 | 68.88 | 55 | 111 | 33.13 | 26.03–40.85 |
| | Jaguapiru | 75 | 31.12 | 12 | 63 | 16.00 | 8.55–26.28 |
| | | | | | | | |
| Ethnicity | | | | | | | |
| | Guarani (Kaiowá) | 178 | 73.86 | 52 | 126 | 29.21 | 22.65–36.48 |
| | Guarani (Nhandeva) | 3 | 1.24 | 1 | 2 | 33.33 | 0.84–90.57 |
| | Terena | 40 | 16.60 | 9 | 31 | 22.50 | 10.84–38.45 |
| | Outros | 20 | 8.30 | 5 | 15 | 25.00 | 08.66–49.10 |
| Education | | | | | | | |
| | Up to 8 years | 200 | 82.98 | 58 | 142 | 29.00 | 22.28–35.82 |
| | Over 9 years | 41 | 17.02 | 9 | 32 | 21.95 | 10.56–37.61 |
| Employment status | | | | | | | |
| | Employed | 46 | 19.09 | 16 | 30 | 34.78 | 21.35–50.25 |
| | Unemployed | 195 | 80.91 | 51 | 144 | 26.15 | 20.14–32.91 |
| Income (USD)/Family Unit | | | | | | | |
| | < 283 | 173 | 71.78 | 53 | 120 | 30.63 | 23.86–38.08 |
| | 283–566 | 60 | 24.90 | 14 | 46 | 23.33 | 13.38–36.04 |
| | 567–1132 | 7 | 2.90 | 0 | 7 | 0 | 0–40.96 |
| | >1133 | 1 | 0.41 | 0 | 1 | 0 | 0–97.50 |
| Government Benefit | | | | | | | |
| | Family Grant | 167 | 69.29 | 53 | 114 | 31.73 | 24.76–39.37 |
| | Retirement | 21 | 8.71 | 5 | 16 | 22.31 | 08.22–47.17 |
| | None | 47 | 19.50 | 9 | 37 | 19.15 | 9.15–33.26 |
| | Do not know | 6 | 2.49 | 0 | 6 | 0 | 0–45.93 |
| **Behavioral characteristics** | | | | | | | |
| Marital status | | | | | | | |
| | Single | 56 | 38.50 | 22 | 34 | 39.29 | 27.58–52.37 |
| | Married | 185 | 61.50 | 45 | 140 | 24.32 | 18.71–30.99 |
| History of cancer | | | | | | | |
| | Yes | 17 | 7.05 | 5 | 12 | 29.41 | 10.31–55.96 |
| | No | 224 | 92.95 | 62 | 162 | 27.68 | 21.93–34.03 |
| Alcohol use | | | | | | | |
| | Yes | 42 | 17.43 | 12 | 30 | 29.57 | 15.72–44.58 |
| | No | 199 | 82.57 | 55 | 144 | 27.64 | 21.55–34.41 |
| Tobacco use | | | | | | | |
| | Yes | 28 | 11.62 | 9 | 19 | 31.14 | 14.88–52.35 |
| | No | 213 | 88.38 | 58 | 155 | 27.23 | 21.37–33.73 |
| Illicit drug use | | | | | | | |
| | Yes | 2 | 0.83 | 2 | 0 | 100.00 | 15.81–100.00 |

*(Continued)*

**Table 2.** (Continued)

| Variable | | N | % | Positive | Negative | TV % | 95% CI* |
|---|---|---|---|---|---|---|---|
| | No | 239 | 99.17 | 65 | 174 | 27.20 | 21.66–33.31 |
| Share objects (syringe and hygiene supplies) | | | | | | | |
| | Yes | 10 | 4.15 | 6 | 4 | 60.00 | 26.24–87.84 |
| | No | 231 | 95.85 | 61 | 170 | 26.41 | 20.84–32.59 |
| Partners in the past 1year | | | | | | | |
| | > one | 203 | 84.23 | 54 | 149 | 26.60 | 22.10–33.07 |
| | < two | 38 | 16.73 | 13 | 25 | 34.21 | 21.21–50.11 |
| **Condom use in the last year** condom use in the last year | | | | | | | |
| All women | | | | | | | |
| | Yes | 26 | 10.79 | 4 | 22 | 15.38 | 4.36–34.87 |
| | No | 215 | 89.21 | 63 | 152 | 29.30 | 23.31–35.88 |
| Married women | | | | | | | |
| | Yes | 16 | 8.64 | 1 | 15 | 6.28 | 0.15–30.23 |
| | No | 169 | 91.35 | 44 | 125 | 26.03 | 19.59–33.33 |
| Single women | | | | | | | |
| | Yes | 11 | 19.64 | 4 | 7 | 36.36 | 10.92–69.20 |
| | No | 45 | 80.36 | 18 | 27 | 40.00 | 25.69–55.66 |
| History of STIs | | | | | | | |
| | Yes | 5 | 2.07 | 3 | 2 | 60.00 | 14.66–94.73 |
| | No | 236 | 97.43 | 64 | 172 | 27.12 | 21.55–33.27 |
| Urethral Discharge | | | | | | | |
| | Yes | 25 | 10.37 | 7 | 18 | 28.00 | 12.07–49.39 |
| | No | 216 | 89.63 | 60 | 156 | 27.70 | 21.92–34.26 |

*Clopper-Pearson exact.

(p = 0.009), the risk is two times among single women (p = 0.043), and sharing objects increases the risk of infection by four times (p = 0.04). No statistically significant association was found with non-use of condoms (p = 0.261) and aged under 30 years (p = 0.1295). However, when combined variables were analyzed, the risk of TV infection increased four times (p = 0.005) in the group of single women aged below 30 years and three times (p = 0.000) in single women who do not use condoms (Table 3).

## Discussion

This is the first study that measures the extent of TV infection among women from an indigenous population in Brazil. Very few existing studies do not have enough data to describe the prevalence and associated factors for trichomoniasis and other STIs in this population [15–17]. Here, we observed a prevalence of 27.8% of the TV infection, which was dependent of laboratory diagnostic technique used [7]. Indeed, PCR (27.8%) and direct microscopy (7.41%) diagnosis showed a considerable difference in the prevalence of TV and might be due to factors, such as bacterial load and test sensitivity [18].

Direct microscopy is frequently used in clinical because it is inexpensive, fast, and non-labor-intensive compared with alternatives such as wet mount preparation, culture, and PCR [19]. In addition, direct microscopy is commonly used in basic health units in Brazil, for the diagnosis of cellular changes, and of pathogenic microorganisms, although with low sensitivity

**Table 3. Distribution of women based on dependence correlation with *T. vaginalis* infection.**

| | X² | *p*-value | OR | 95% CI | *p*-value |
|---|---|---|---|---|---|
| **Socioeconomic characteristics** | | | | | |
| Age < 30 | 3.89 | 0.030 | 1.63 | 0.91–2.93 | 0.129 |
| Village Bororó | 7.55 | 0.004 | 2.60 | 1.29–5.22 | 0.001 |
| Schooling less than 8 years | 0.84 | 0.237 | 1.45 | 0.65–3.23 | 0.467 |
| Income less than 283.00USD | 2.45 | 0.078 | 1.70 | 0.87–3.33 | 0.159 |
| Government income dependent | 4.197 | 0.027 | 2.18 | 0.99–4.76 | 0.069 |
| **Behavioral characteristics** | | | | | |
| Single women | 4.79 | 0.023 | 2.01 | 1.07–3.79 | 0.043 |
| Share objects | 5.39 | 0.030 | 4.18 | 1.14–15.31 | 0.050 |
| Do not use a condom | 1.30 | 0.181 | 2.28 | 0.75–6.88 | 0.261 |
| Singles under 30 years | 9.31 | 0.004 | 4.08 | 1.56–10.64 | 0.005 |
| Single and not using condom | 13.56 | 0.000 | 3.11 | 1.67–5.78 | 0.000 |

[20], especially in women who do not have symptoms [21]. However, a disadvantage of this convenience is the lower detection rate of direct microscopy relative to PCR as observed in our study (3.75 lower). Taking into account that most cases of TV are asymptomatic or have common symptoms among uninfected patients [19, 22], increasing even more the number of underdiagnosis cases of the disease [10], it is imperative to carry out population screening and to implement information measures on the disease as well as more specific and precise diagnostic methods [19, 22].

The results of this study are similar to the already published ones where the higher prevalence of TV is found in a highly vulnerable population: people deprived of their liberty, sex workers, and populations with low socioeconomic indexes [2, 4, 8, 9]. In this study, 71% of the women were observed to have a family income of less than 189.00 USD, which represents less than 20% of the average family income from the rest of Brazil [19, 22]. The high rates of unemployed women (80.91%), women dependent on government benefits (79.9%), or with low education (less than eight years) (83%), corroborate with other data that indicate the low socioeconomic level of this population [20, 23].

Several demographic and behavioral factors have been associated with TV infection. Our study did not show any association between level of education and family income, but an association was observed in women who depended on government benefits. Also, an association between living in the Bororó village (33.13%) and TV infection was observed, against 16% in the Jaguapiru village, showed that the risk of TV infection increased by 2.5 times in women living in the Bororó village. Data already described in the literature show that the Bororó has a lower socioeconomic index than Jaguapiru village [23], and our study found that only 13% of the women living in the Bororó village had economic activity, while this percentage rises to 32% for women living in the Jaguapiru village. Those findings might be related with the low socioeconomic index, but more studies are necessary to determine if any other factors can be involved.

We also observed that the use of condoms has a low association with TV infection in this population but is important to report that among women said they did not use condoms, 78% were married women. Considering the prevalence among the group of the married women who do not use condoms, the prevalence of TV was 26.03% and among married women who use condoms the prevalence was 6.25%. Besides that, for single women, the risk of TV infection increases by two times and a strong relationship between TV infection and single women who do not use condoms is observed. A prevalence rate of 35.80% was found in women aged

below 30 years and TV infection; this number becomes higher (57.89%) when stratified women aged below 30 years and single.

Concomitant infection between TV and *G. vaginalis* and TV and *C. albicans* were found in 3 women, respectively. The concomitant infection is commonly found in cases where TV makes the vaginal environment susceptible to infection by *C. albicans*, and other microorganisms, such as *G. vaginalis* joins leading to the development of bacterial vaginosis (BV) [24–30]. The presence of atypical squamous cells of unknown significance was observed in 10 of the sample population, and HPV characteristic lesions were observed in 14 women, and among them, 5/24 of women who show lesion on the pap smear, were infected concomitantly with TV. Although HPV has not been diagnosed, the injuries indicates the presence of this viral infection [28, 31]. Some studies have investigated the relationship between BV and STIs, as well as ASC-US and other changes found in the pap smear [28, 32]. These data suggest that the overall prevalence of TV in the indigenous women evaluated may be much higher than the 27.8% found in this study.

There are a few limitations in our study. The participants were enrolled from Basic Health Units of the two indigenous reserves, which affects the generalizability of the results, because sampling is convenient, and thus possible biases may occur, for example, women who had a better chance of accessing the basic health unit, women who are more likely to seek a health service or also women who had some symptom. Sample distribution was unequal between the two reserves, and we must be careful about establishing comparisons. Another limitation is related to variables obtained through the questionnaire, once participants were reluctant to answer some questions, mainly those related to sexual behavior and the use of alcohol and illegal drugs, and thus introducing reporting biases.

In conclusion, a high prevalence of TV infection in indigenous women from two indigenous reserves in the Midwest region of Brazil was observed in this study. Certain groups such as (i) women dependent on government benefits, (ii) single women aged below 30 years who do not use condoms were most likely to have TV infection. In addition, TV infection is underdiagnosis in this population, therefore, improved STI surveillance, with the use of more accurate diagnostic methods, as well as proper management of the risk factors in different indigenous population groups, is vital for health promotion and prevention efforts.

## Supporting information

**S1 Checklist. STROBE statement—checklist of items that should be included in reports of *cross-sectional studies*.**
(PDF)

## Acknowledgments

We thank the team of nurses from the basic health units of the Dourados/MS indigenous reserve for their support during the sample collection. We also appreciate the support of the Government of the State of Mato Grosso do Sul and, Health Department of the state of Mato Grosso do Sul.

## Author Contributions

**Conceptualization:** Marcelo dos Santos Barbosa, Iara Beatriz Andrade de Souza, Simone Simionatto, Silvana Beutinger Marchioro.

**Data curation:** Marcelo dos Santos Barbosa, Simone Simionatto, Silvana Beutinger Marchioro.

**Formal analysis:** Marcelo dos Santos Barbosa, Iara Beatriz Andrade de Souza, Erica Cristina dos Santos Schnaufer, Liliane Ferreira da Silva, Crhistinne Carvalho Maymone Gonçalves, Simone Simionatto, Silvana Beutinger Marchioro.

**Funding acquisition:** Crhistinne Carvalho Maymone Gonçalves, Simone Simionatto, Silvana Beutinger Marchioro.

**Investigation:** Marcelo dos Santos Barbosa, Iara Beatriz Andrade de Souza, Erica Cristina dos Santos Schnaufer, Liliane Ferreira da Silva, Crhistinne Carvalho Maymone Gonçalves, Simone Simionatto, Silvana Beutinger Marchioro.

**Methodology:** Marcelo dos Santos Barbosa, Iara Beatriz Andrade de Souza, Simone Simionatto, Silvana Beutinger Marchioro.

**Project administration:** Marcelo dos Santos Barbosa, Crhistinne Carvalho Maymone Gonçalves, Simone Simionatto, Silvana Beutinger Marchioro.

**Supervision:** Silvana Beutinger Marchioro.

**Writing – original draft:** Marcelo dos Santos Barbosa, Iara Beatriz Andrade de Souza, Erica Cristina dos Santos Schnaufer, Liliane Ferreira da Silva, Crhistinne Carvalho Maymone Gonçalves, Simone Simionatto, Silvana Beutinger Marchioro.

**Writing – review & editing:** Crhistinne Carvalho Maymone Gonçalves, Simone Simionatto, Silvana Beutinger Marchioro.

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
