## [Decision Letter · Decision Letter 0]

5 Aug 2020

PONE-D-20-17319

PREVALENCE OF SEXUALLY TRANSMITTED INFECTIONS IN INDIGENOUS BRAZILIAN WOMEN: WATCH OUT FOR Trichomonas vaginalis INFECTION

PLOS ONE

Dear Dr. Marchioro,

Thank you for submitting your manuscript to PLOS ONE. After careful consideration, we feel that it has merit but does not fully meet PLOS ONE’s publication criteria as it currently stands. Therefore, we invite you to submit a revised version of the manuscript that addresses the points raised during the review process.

We look forward to receiving your revised manuscript.

Kind regards,

Catherine E Oldenburg

Academic Editor

PLOS ONE

Journal Requirements:

Reviewers' comments:

Reviewer's Responses to Questions

**Comments to the Author**

1. Is the manuscript technically sound, and do the data support the conclusions?

Reviewer #1: Yes

Reviewer #2: Yes

2. Has the statistical analysis been performed appropriately and rigorously? 

Reviewer #1: Yes

Reviewer #2: Yes

3. Have the authors made all data underlying the findings in their manuscript fully available?

Reviewer #1: Yes

Reviewer #2: Yes

4. Is the manuscript presented in an intelligible fashion and written in standard English?

Reviewer #1: Yes

Reviewer #2: Yes

5. Review Comments to the Author

Reviewer #1: This is a very good article, with relevant data from a population in which little is known. The data presented here have great relevance in the general context, as it is a gap in the literature and evidence knowledge that tends to generate actions in the field of public health. The article is well structured and meets the steps foreseen for a scientific article. All methodological steps were well described, identifying all the steps performed in this study.

However, there is a problem with the objectives proposed in the article. The purpose of the study does not correspond to what is developed in the body of the article. There is a focus throughout the article, from the introduction, analysis, and discussion, on the infection by trichomoniasis, but in the objectives, it informs to evaluate also other STIs, mainly viral hepatitis, HIV, and syphilis. I believe that an important decision needs to be made: better to structure and discuss the data in the light of the other sexually transmitted infections presented in the objective, or focus on the discussion in the analysis of the findings of Trichomonas vaginalis infection. The latter seems to be more viable and close to the one presented so far. Maintaining this analysis alone does not make the work of little relevance.

In addition to these indications, I present some specific considerations. Follow below:

1. Title - I didn't understand why part of the title is in capital letters and part is not.

ABSTRACT

1. “The prevalence of the 3.73% for T. pallidum and 0.41% for HIV were found in rapid tests.” Note that the prevalence of viral hepatitis was equal to 0. It is also an important fact.

2. Identify that it is a cross-sectional study and the period that the data collection occurred - "A cross-sectional study was 71 conducted between January and December 2018" was only mentioned in the article´s methodology.

3. "The prevalence of STIs, mainly TV, is among the highest in the world". About what? Among women? Compared to what other infections?

INTRODUCTION

1. “The main and most prevalent STIs described in the literature are caused by Chlamydia 42 trachomatis, Neisseria gonorrhoeae, Trichomonas vaginalis (TV), and Treponema pallidum.” This information is very important. What is the reference? Where is this described?

2. “Trichomoniasis is the most common STI caused by TV.” The information is incomplete. Where? Between who? What is the reference?

3. The focus of the introduction is on trichomoniasis. It scarcely reports on other STIs, especially viral hepatitis, HIV, and syphilis, as indicated in the study's objective.

METHODOLOGY

1. “Considering an expected prevalence of 15%” - Prevalence of what? Line 74

2. “Alere DetermineTM 118 Syphilis TP rapid test was used for the diagnosis of syphilis.” It is important to describe what this TR Identifies. Do you investigate antibodies? Do you identify if you have active (acute) syphilis or the presence of antibodies in the body (may indicate a serological scar)?

RESULT

1. “USD 283.00” - It´s important to contextualize this information for Brazilians. How was this variable categorized? What is the criterion used for this categorization? What does this mean in the Brazilian context in the period studied? Is it the minimum wage?

2. “human 156 papillomavirus (HPV) infected lesions” - was described in the results without commenting on the summary, methodology, or introduction. It is important to contextualize and present all the data that will be presented in the Results section, at least in the methodology.

3. Present the sociodemographic and behavioral variables in the methodology, describing in detail each variable and its categories.

TABLES

1. “Table 2. Distribution of indigenous women based on socioeconomic and behavioral 172 variables” - Is the prevalence of TV not described in this table? In the title, you should mention that too.

2. “Share objects” - What objects are shared?

3. “Condom use” - In which relationships? With a steady partner? With a casual partner? In what period in life? In the last 6 m? Partners in the past 1year “> one and <two” does="" mean="" that="" what="">

DISCUSSION

1. The discussion and analysis of associated factors are restricted to TV infection. The other STIs are not even mentioned. The work is actually about TV.

2. “This difference is might be due to factors, such as bacterial load and test sensitivity [12]. These factors contribute to an underdiagnosis of the disease; once direct microscopy is the only clinical diagnostic method applied [4]. ” - This finding is extremely relevant and should be better discussed. Including, to envision the impact of this in the planning in the actions in the health services.

3. “This data indicate that prevalence rates of infection increases due to socioeconomic factors in this population.” - Line 213-214. The difference in prevalence in these two regions can be explained by the difference in socioeconomic development, but from the data presented we cannot say that it is due to this. Other factors can determine this discrepancy, an example of the community viral load and STI prevalence.

4. “On the other hand, for single women, the risk increases by two times (p = 0.043) and, a strong relationship between TV infection and single women who do not use condoms (p = 0.000) is observed, increasing the risk of having TV infection (p = 0.000) by three times. ” - Line 216-219. This data should be better developed and explained. Condom use can be an important factor in the association between relationship and infection. Discuss this association better.

5. It´s important to discuss better that it is a sampling due to convenience and the bias caused by the study population comes from a health service.

6. In the important conclusion, reinforce the use of more sensitive methods for the diagnosis of VT, considering that there is a discrepancy when comparing the methods of analysis. This is an important finding in this study.</two”>

Reviewer #2: The manuscript is well written and the experimental design is adequate to the proposed objectives. The results of the study point to the high prevalence of STI, especially T. vaginalis, in the indigenous population of a region of Brazil and, although the sample size is small, it demonstrates the vulnerability of this specific population to STIs. However, the text needs some adjustments to be published.

1. In the abstract, the authors state in the methods that they will make rapid tests for hepatitis B and C but do not report any information in the results. In addition, the authors should draw some conclusions about the comparison of molecular and T. vaginalis microscopy tests.

2. In the methodology, it would be interesting if in the questionnaire the question about sexual partners included indigenous people and other people outside the community.

The authors must inform how the quality control of the DNA extraction of the samples was carried out. The DNA extracted from the culture of ATCC 30236 strain was used as a positive control of the reaction, but we need to make sure that the negative samples had viable DNA. An internal control of the reaction would be interesting, with the amplification of a human gene to guarantee the result.

The authors cite the collection of Vaginal samples (line 91) for PCR and Direct microscopic examination. They do not mention blood collection for the performance of rapid tests. Authors should add this information.

In line 116, the authors put Federal University of Espírito Santo, Vitória, Brazil after citing the rapid HIV test. Does this mean that the test was the test manufacturer?

At the end of the methodology, the authors do not mention which test was performed to detect anti-HCV.

The authors should mention in the methodology which procedures were adopted for the women in the study when the studied infections occurred

3. In results, on line 140, the authors claim that 73.86% were from the Guarani-Kaiowá ethnic group. And the rest of the group? Authors must inform ethnicities in the methodology, when mentioning Bororó and Jaguapiru village (line 79)

In line 149, hepatitis C is again cited. Authors should add hepatitis C to other topics if they want to continue with the information.

Table 1 mentions Cytologic change, however, this information is not included in any previous topic, neither in the abstract nor in the methodology. Authors should review this.

Table 1 is confused in the "n in TV PCR" column, as it concerns only T. vaginalis and in the table several other agents are mentioned. In my opinion, this description can be made in the text, without requiring a table representation. The percentage of 25.37% in T. vaginalis in Direct Microscopy was confused and has no reference in the text.

The authors do not mention the observation of C. albicans (line 155) in the abstract, nor do they mention in the methodology that they would also perform cytology in the collected cervical samples, to justify the finding of possible injury by HPV and ASC-US.

In table 2, the authors cite Ethnicity. This information must also be included in the methodology. In addition, in the table it would be interesting to detail "others" (which is written in Portuguese), because in the Guarani-Nhandeva ethnic group only 3 individuals are mentioned.

In table 2, the authors must adjust the decimal places of all percentage results.

6. PLOS authors have the option to publish the peer review history of their article (what does this mean?). If published, this will include your full peer review and any attached files.

Reviewer #1: No

Reviewer #2: No

---

## [Author Response · Author response to Decision Letter 0]

26 Aug 2020

Revision note

August 26th, 2020.

Dr. Joerg Heber 

PLOS ONE - Editor-in-Chief

RE: PONE-D-20-17319R1

Dear Editor

 Thank you very much for inviting us to submit a revised version of our manuscript entitled: “Prevalence of sexually transmitted infections in indigenous Brazilian women: watch out for Trichomonas vaginalis infection (PONE-D-20-17319)” for publication in Plos One. Indeed, the reviewers have raised a number of important concerns. We have now made a through revision of the manuscript taking into account these points, which helped in improving the quality of the manuscript. Please find bellow a point-by-point response to the reviewer’s and to your comments.

Reviewer #1: 

This is a very good article, with relevant data from a population in which little is known. The data presented here have great relevance in the general context, as it is a gap in the literature and evidence knowledge that tends to generate actions in the field of public health. The article is well structured and meets the steps foreseen for a scientific article. All methodological steps were well described, identifying all the steps performed in this study.

However, there is a problem with the objectives proposed in the article. The purpose of the study does not correspond to what is developed in the body of the article. There is a focus throughout the article, from the introduction, analysis, and discussion, on the infection by trichomoniasis, but in the objectives, it informs to evaluate also other STIs, mainly viral hepatitis, HIV, and syphilis. I believe that an important decision needs to be made: better to structure and discuss the data in the light of the other sexually transmitted infections presented in the objective, or focus on the discussion in the analysis of the findings of Trichomonas vaginalis infection. The latter seems to be more viable and close to the one presented so far. Maintaining this analysis alone does not make the work of little relevance.

In addition to these indications, I present some specific considerations. Follow below:

TITLE

1. I didn't understand why part of the title is in capital letters and part is not.

R: We have corrected the existing error and adapted the title to a version of the work in which we specifically deal with TV.

“Prevalence and factors associated with Trichomonas vaginalis infection in indigenous Brazilian women” (P1/L1)

ABSTRACT

1. “The prevalence of the 3.73% for T. pallidum and 0.41% for HIV were found in rapid tests.” Note that the prevalence of viral hepatitis was equal to 0. It is also an important fact.

R: We received the comment about the possibility of maintaining the work only on TV, we accepted this suggestion and reformatted the document keeping only the data on T. vaginalis and microscopic findings.

2. Identify that it is a cross-sectional study and the period that the data collection occurred - "A cross-sectional study was 71 conducted between January and December 2018" was only mentioned in the article´s methodology.

R: We corrected the absence of the information.

“We conducted a cross-sectional study between January and December 2018,” (P2/L17)

3. "The prevalence of STIs, mainly TV, is among the highest in the world". About what? Among women? Compared to what other infections?

R: We added the information in order to supply the suggestion.

“The high prevalence TV found in this population is comparable to highly vulnerable populations, as prisoners, sex workers and women in regions with low socioeconomic levels, moreover, seems to be an underdiagnosis for this infection”. (P2/L31)

INTRODUCTION

1. “The main and most prevalent STIs described in the literature are caused by Chlamydia 42 trachomatis, Neisseria gonorrhoeae, Trichomonas vaginalis (TV), and Treponema pallidum.” This information is very important. What is the reference? Where is this described?

R: We corrected the absence of reference.

“Rowley J, Vander Hoorn S, Korenromp E, Low N, Unemo M, Abu-Raddad LJ, et al. Chlamydia, gonorrhoea, trichomoniasis and syphilis: global prevalence and incidence estimates, 2016. Bull World Health Organ [Internet]. 2019 Aug 1 [cited 2020 Jan 24];97(8):548-562P.”

“WHO. Global health sector strategy on sexually transmitted infections 2016–2021. Towards ending STIs. Report. Geneva: 2016 June. Report No. WHO/RHR/16.09; 2016.”

2. “Trichomoniasis is the most common STI caused by TV.” The information is incomplete. Where? Between who? What is the reference?

R: There was an inconsistency in the writing that was corrected.

“Trichomoniasis, caused by TV, is the non-viral STI most common in the world [3, 4]” (P3/L43)

3. The focus of the introduction is on trichomoniasis. It scarcely reports on other STIs, especially viral hepatitis, HIV, and syphilis, as indicated in the study's objective.

R: We accepted this suggestion and reformatted the document, keeping only the data on TV and microscopic findings.

METHODOLOGY

1. “Considering an expected prevalence of 15%” - Prevalence of what? Line 74

R: We corrected the inconsistency in the text.

“Considering an average prevalence of T. vaginallis of 15% (4–27%)” (P4/L79)

2. “Alere DetermineTM 118 Syphilis TP rapid test was used for the diagnosis of syphilis.” It is important to describe what this TR Identifies. Do you investigate antibodies? Do you identify if you have active (acute) syphilis or the presence of antibodies in the body (may indicate a serological scar)?

R: We removed information about other STIs from the document and kept only information about TV and microscopic findings.

RESULT

1. “USD 283.00” - It´s important to contextualize this information for Brazilians. How was this variable categorized? What is the criterion used for this categorization? What does this mean in the Brazilian context in the period studied? Is it the minimum wage?

R: We consider the comment and add the information.

“Family with income below USD 283.00 (which corresponds to minimum Brazilian wage)” (P7/L148)

2. “human 156 papillomavirus (HPV) infected lesions” - was described in the results without commenting on the summary, methodology, or introduction. It is important to contextualize and present all the data that will be presented in the Results section, at least in the methodology.

R: We added information in the summary and introduction in order to remedy the deficiency in the text.

“Direct microcopy also found 21 (8.71%) and 8 (3,31%) women were infected with Gardnerella vaginalis and Candida albicans, respectively. In addition, 10 women presented atypical squamous cells of unknown significance and 14 lesions suggestive of HPV.” (P1/L27)

“The diagnosis of TV in the public health network is basically performed through direct microscopy of a sample obtained in the Pap smear test. Although the Pap smear has specificity for T. vaginalis, there are sensitivity limitations that prevent the use of this technique for the diagnosis of infection [3, 5]. The main focus of this technique is on the search for cellular changes such as atypical granular cells and squamous lesions, other findings such as Gardnerella vaginallis infection or Candida albicans, end up being in the background, which can also be related with problems in the female reproductive tract, especially the imbalance of bacterial flora, increasing the risk of infection by other STIs. [6, 7, 8]” (P2/L49)

3. Present the sociodemographic and behavioral variables in the methodology, describing in detail each variable and its categories.

R: Variables were added to the text.

“A questionnaire addressing demographic was given to the participants contemplating the following variables: village (Bororó and Jaguapirú), and ethnicities (Guarani-Kaiowa, Guarani Nhandeva, Terena, and others not predominant in the region); socioeconomic data (age, employment status, Income, government benefits, and years of schooling); clinical data: history of cancer, symptoms, and other STIs diagnoses; and behavioral factors: marital status, number of partners in the last year, alcohol use, tobacco use, illicit drug use, share objects (Syringes and personal hygiene supplies), condom use. Categorical variables were represented as “Yes” or “No”, and numerical variables were categorized later.” (P5/L90)

TABLES

1. “Table 2. Distribution of indigenous women based on socioeconomic and behavioral 172 variables” - Is the prevalence of TV not described in this table? In the title, you should mention that too.

R: We fixed the error in the table title.

“Prevalence of T. vaginalis in indigenous women in relation to sociodemographic and behavioral variables” (P9/L184)

2. “Share objects” - What objects are shared?

R: were added to the text.

“Syringes and personal hygiene supplies” (Table 2)

3. “Condom use” - In which relationships? With a steady partner? With a casual partner? In what period in life? In the last 6 m? Partners in the past 1year “> one and

R: We reformulated the paragraph in order to better describe the relationship between condom use and TV infection in this population, we also formatted table 1 to include the requested information.

DISCUSSION

1. The discussion and analysis of associated factors are restricted to TV infection. The other STIs are not even mentioned. The work is actually about TV.

R: We changed the text to keep only Infection by TV, since the information about other infections does not include enough numbers to confidently describe the risk factors.

2. “This difference is might be due to factors, such as bacterial load and test sensitivity [12]. These factors contribute to an underdiagnosis of the disease; once direct microscopy is the only clinical diagnostic method applied [4]. ” - This finding is extremely relevant and should be better discussed. Including, to envision the impact of this in the planning in the actions in the health services.

R: We discussed the result better. 

“Direct microscopy is frequently used in clinical because it is inexpensive, fast, and non-labor-intensive compared with alternatives such as wet mount preparation, culture, and PCR [19]. In addition, direct microscopy is commonly used in basic health units in Brazil, for the diagnosis of cellular changes, and of pathogenic microorganisms, although with low sensitivity [20], especially in women who do not have symptoms [21]. However, a disadvantage of this convenience is the lower detection rate of direct microscopy relative to PCR as observed in our study (3.75 lower). Taking in to account that most cases of TV are asymptomatic or have common symptoms among uninfected patients [19,22], increasing even more the number of underdiagnosis cases of the disease [10], it is imperative to carry out population screening and to implement information measures on the disease as well as more specific and precise diagnostic methods [19,22].” (P12/L205)

3. “This data indicate that prevalence rates of infection increase due to socioeconomic factors in this population.” - Line 213-214. The difference in prevalence in these two regions can be explained by the difference in socioeconomic development, but from the data presented we cannot say that it is due to this. Other factors can determine this discrepancy, an example of the community viral load and STI prevalence.

R: We reformulated the statement, making it clear that this possibility can be explained by other factors not covered in the study.

“Those findings might be related with the low socioeconomic index, but more studies are necessary to determine if any other factors can be involved.” (P13/L234)

4. “On the other hand, for single women, the risk increases by two times (p = 0.043) and, a strong relationship between TV infection and single women who do not use condoms (p = 0.000) is observed, increasing the risk of having TV infection (p = 0.000) by three times. ” - Line 216-219. This data should be better developed and explained. Condom use can be an important factor in the association between relationship and infection. Discuss this association better.

R: We reformulated the paragraph in order to better describe the relationship between condom use and TV infection in this population, we also formatted table 1 to include the requested information.

“We also observed that the use of condoms has a low association with TV infection in this population (p = 0.261) but is important to report that among women said they did not use condoms, 78% were married women. Considering the prevalence among the group of the married women who do not use condoms, the prevalence of TV was 26.03%, among married women who use condoms the prevalence was 6.25% (Table 2). Besides that, for single women, the risk of TV infection increases by two times (p = 0.043), and a strong relationship between TV infection and single women who do not use condoms (p = 0.000) is observed. A prevalence rate of 35.80% was found in women aged below 30 years and TV infection (p = 0.03); this number becomes higher when stratified women aged below 30 years and single (p = 0.04) (Table 3).” (P13/L236)

 5. It´s important to discuss better that it is a sampling due to convenience and the bias caused by the study population comes from a health service.

R: We insert the information in the text.

“because sampling is convenient, and thus possible biases may occur, for example, women who had a better chance of accessing the basic health unit, women who are more likely to seek a health service or also women who had some symptom.” (P14/L260)

6. In the important conclusion, reinforce the use of more sensitive methods for the diagnosis of VT, considering that there is a discrepancy when comparing the methods of analysis. This is an important finding in this study.

R: We insert the information in the text.

“TV infection is underdiagnosis in this population, therefore, improved STI surveillance, with the use of more accurate diagnostic methods,” (P14/L270)

Reviewer #2: 

The manuscript is well written, and the experimental design is adequate to the proposed objectives. The results of the study point to the high prevalence of STI, especially T. vaginalis, in the indigenous population of a region of Brazil and, although the sample size is small, it demonstrates the vulnerability of this specific population to STIs. However, the text needs some adjustments to be published.

R: Dear reviewer, some of the issues raised are related to STIs other than TV, the other reviewer indicated the possibility of making the document more objective, removing this data and maintaining only TV. We chose to follow this suggestion. In this sense, the questions raised about the other STIs were removed from the document.

1. In the abstract, the authors state in the methods that they will make rapid tests for hepatitis B and C but do not report any information in the results. In addition, the authors should draw some conclusions about the comparison of molecular and T. vaginalis microscopy tests.

R: We inserted the information related to the comparison between the molecular test and microscopy. The information on other STIs were removed from the document, following suggestion from the reviewer 1.

“Therefore, a routine test program, as well as a review of the diagnostic method used, is encouraged for proper management.” (P2/L33)

2. In the methodology, it would be interesting if in the questionnaire the question about sexual partners included indigenous people and other people outside the community.

R: We accepted the suggestion rose by the reviewer, it really would be something of great value for the work, but unfortunately the issue was not addressed, as it could bring some constraints to the participants and they could deny the enrolment in the study.

The authors must inform how the quality control of the DNA extraction of the samples was carried out. The DNA extracted from the culture of ATCC 30236 strain was used as a positive control of the reaction, but we need to make sure that the negative samples had viable DNA. An internal control of the reaction would be interesting, with the amplification of a human gene to guarantee the result.

R: We carry out the quality control of the samples through spectrophotometry, agarose gel of the extracted samples and by PCR for β-globin gene. Unfortunately, the information’s were missed in the first version from the manuscript, but were added as suggested by the reviewer

 “DNA was quantified using the UV-Vis Biodrop µLite Spectrophotometer (Biochrom, USA). DNA integrity was evaluated by agarose gel and by amplification of β-globin gene (268 base pairs) representing the internal control of the reaction.” (P5/L111)

The authors cite the collection of Vaginal samples (line 91) for PCR and Direct microscopic examination. They do not mention blood collection for the performance of rapid tests. Authors should add this information.

R: We follow the suggestion from the reviewer 1 and removed all the informations about other STIs from the text, thus keeping the work focused only on TV infection.

In line 116, the authors put Federal University of Espírito Santo, Vitória, Brazil after citing the rapid HIV test. Does this mean that the test was the test manufacturer?

R: Yes, it deals with the place where the test was produced, there was a small flaw in the writing. But we decided to remove information about HIV and other STIs from the manuscript.

At the end of the methodology, the authors do not mention which test was performed to detect anti-HCV.

R: In the same way as the previous questions, we remove this information, keeping only TV infection as the focus of the manuscript.

The authors should mention in the methodology which procedures were adopted for the women in the study when the studied infections occurred

R: Our study had a partnership with the basic health units and all the information obtained were summitted to the health unit of the units, where appropriate procedures were carried out. We also add this information in the text.

“The exams results were submitted to the Health Unit, where the necessary treatment measures could be taken for the patients in whom an infection was detected.” (P5/L102)

3. In results, on line 140, the authors claim that 73.86% were from the Guarani-Kaiowá ethnic group. And the rest of the group? Authors must inform ethnicities in the methodology, when mentioning Bororó and Jaguapiru village (line 79)

R: We include ethnicities in the methodology.

“Data and samples were obtained during their visit to the Basic Health Units of the two indigenous reserves evaluated: Bororó and Jaguapiru village, where Guarani (Kaiowá and Nhandeva) and Terena ethnicities predominate. (P4/L87)

In line 149, hepatitis C is again cited. Authors should add hepatitis C to other topics if they want to continue with the information.

R: We removed data about other STIs from the manuscript.

Table 1 mentions Cytologic change, however, this information is not included in any previous topic, neither in the abstract nor in the methodology. Authors should review this.

R: We added information in the summary and introduction in order to correct this issue..

“Direct microcopy also found 21 (8.71%) and 8 (3,31%) women were infected with Gardnerella vaginalis and Candida albicans, respectively. In addition, 10 women presented atypical squamous cells of unknown significance and 14 lesions suggestive of HPV.” (P1/L27)

“The diagnosis of TV in the public health network is basically performed through direct microscopy of a sample obtained in the Pap smear test. Although the Pap smear has specificity for T. vaginalis, there are sensitivity limitations that prevent the use of this technique for the diagnosis of infection [3, 5]. The main focus of this technique is on the search for cellular changes such as atypical granular cells and squamous lesions, other findings such as Gardnerella vaginallis infection or Candida albicans, end up being in the background, which can also be related with problems in the female reproductive tract, especially the imbalance of bacterial flora, increasing the risk of infection by other STIs. [6, 7, 8]” (P2/L49)

Table 1 is confused in the "n in TV PCR" column, as it concerns only T. vaginalis and in the table several other agents are mentioned. In my opinion, this description can be made in the text, without requiring a table representation. The percentage of 25.37% in T. vaginalis in Direct Microscopy was confused and has no reference in the text.

R: We removed the information that was causing confusion in table 1.

The authors do not mention the observation of C. albicans (line 155) in the abstract, nor do they mention in the methodology that they would also perform cytology in the collected cervical samples, to justify the finding of possible injury by HPV and ASC-US.

R: We added information in the summary and introduction to correct this issue in the manuscript.

“Direct microcopy also found 21 (8.71%) and 8 (3,31%) women were infected with Gardnerella vaginalis and Candida albicans, respectively. In addition, 10 women presented atypical squamous cells of unknown significance and 14 lesions suggestive of HPV.” (P1/L27)

“The diagnosis of TV in the public health network is basically performed through direct microscopy of a sample obtained in the Pap smear test. Although the Pap smear has specificity for T. vaginalis, there are sensitivity limitations that prevent the use of this technique for the diagnosis of infection [3, 5]. The main focus of this technique is on the search for cellular changes such as atypical granular cells and squamous lesions, other findings such as Gardnerella vaginallis infection or Candida albicans, end up being in the background, which can also be related with problems in the female reproductive tract, especially the imbalance of bacterial flora, increasing the risk of infection by other STIs. [6, 7, 8]” (P2/L49)

In table 2, the authors cite Ethnicity. This information must also be included in the methodology. In addition, in the table it would be interesting to detail "others" (which is written in Portuguese), because in the Guarani-Nhandeva ethnic group only 3 individuals are mentioned.

R: In the study design, the ethnic groups that are predominant were considered to be the Guarani and Terena ethnic groups, the Guarani ethnic group, in turn, is divided into two subgroups, determined Kaiowa and Nhandewa, during the questionnaire this subgroup was divided, but most of the patients declared themselves Guarani-Kaiowa. With regard to the determined group "others", these are groups that are considered to have a lower proportion within the population. This proportion was described in the methodology according to the request.

“A questionnaire addressing demographic was given to the participants contemplating the following variables: village (Bororó and Jaguapirú), and ethnicities (Guarani-Kaiowa, Guarani Nhandeva, Terena, and others not predominant in the region); socioeconomic data (age, employment status, income, government benefits, and years of schooling); clinical data: history of cancer, symptoms, and other STIs diagnoses; and behavioral factors: marital status, number of partners in the last year, alcohol use, tobacco use, illicit drug use, share objects (syringes and personal hygiene supplies), condom use. Categorical variables were represented as “yes” or “no”, and numerical variables were categorized later.” (P5/L90)

In table 2, the authors must adjust the decimal places of all percentage results.

R: It has been corrected and adjusted.

We hope that the manuscript can be now reconsidered for publication in Plos One. Please, let me know if any further change in the manuscript is required.

Yours Sincerely,

Silvana B. Marchioro (corresponding author)

DVM, PhD

---

## [Decision Letter · Decision Letter 1]

15 Sep 2020

PONE-D-20-17319R1

Prevalence and factors associated with Trichomonas vaginalis infection in indigenous Brazilian women

PLOS ONE

Dear Dr. Marchioro,

Thank you for submitting your manuscript to PLOS ONE. After careful consideration, we feel that it has merit but does not fully meet PLOS ONE’s publication criteria as it currently stands. Therefore, we invite you to submit a revised version of the manuscript that addresses the points raised during the review process.

We look forward to receiving your revised manuscript.

Kind regards,

Catherine E Oldenburg

Academic Editor

PLOS ONE

Additional Editor Comments (if provided):

Thank you for addressing the reviewers' comments. Reviewer 1 has two additional minor suggestions for the article. 

Reviewers' comments:

Reviewer's Responses to Questions

**Comments to the Author**

1. If the authors have adequately addressed your comments raised in a previous round of review and you feel that this manuscript is now acceptable for publication, you may indicate that here to bypass the “Comments to the Author” section, enter your conflict of interest statement in the “Confidential to Editor” section, and submit your "Accept" recommendation.

Reviewer #1: All comments have been addressed

Reviewer #2: All comments have been addressed

2. Is the manuscript technically sound, and do the data support the conclusions?

Reviewer #1: Yes

Reviewer #2: Yes

3. Has the statistical analysis been performed appropriately and rigorously? 

Reviewer #1: Yes

Reviewer #2: Yes

4. Have the authors made all data underlying the findings in their manuscript fully available?

Reviewer #1: Yes

Reviewer #2: Yes

5. Is the manuscript presented in an intelligible fashion and written in standard English?

Reviewer #1: Yes

Reviewer #2: Yes

6. Review Comments to the Author

Reviewer #1: The article presented here is better structured and discussed. It is indicated for publication, according to my assessment. The indications made in the first review were corrected satisfactorily, and I don´t have any more major requests.

I have only two indications to do. The title of figure 1 (Fig 1. A sampling flowchart showing the number of samples collected and diagnostic methods performed) is found in the middle of the article (line 152-153). This must appear immediately before the figure, and not alone in the middle of the text. The authors return the results data in the discussion, once again, the indication of the tables. I believe that this indication of the tables in the discussion is not necessary (lines 234 and 239), given that it was made in the results.

This is an article relevant to the field of public health and the data here exposed are well presented. I take this opportunity to thank you for the invitation and opportunity to review this work. Nothing more to say.

Kind regards,

F. S.

Reviewer #2: (No Response)

7. PLOS authors have the option to publish the peer review history of their article (what does this mean?). If published, this will include your full peer review and any attached files.

Reviewer #1: No

Reviewer #2: No

---

## [Author Response · Author response to Decision Letter 1]

17 Sep 2020

Revision Note

September 17th, 2020

Dr. Catherine E Oldenburg

Academic Editor

PLOS ONE

Dear Editor

 Thank you very much for inviting us to submit a revised version of our manuscript entitled: “Prevalence and factors associated with Trichomonas vaginalis infection in indigenous Brazilian women (PONE-D-20-17319R1)” for publication in PLoS One. We reviewed the points raised by reviewer 1 in this second review. Please find bellow a point-by-point response to the reviewer’s and to your comments.

We hope that the manuscript can be now reconsidered for publication in PLos One. Please, let me know if any further change in the manuscript is required.

Yours Sincerely,

Silvana B. Marchioro (corresponding author)

DVM, PhD

Comments to the Author

Reviewer #1: The article presented here is better structured and discussed. It is indicated for publication, according to my assessment. The indications made in the first review were corrected satisfactorily, and I don´t have any more major requests.

I have only two indications to do. The title of figure 1 (Fig 1. A sampling flowchart showing the number of samples collected and diagnostic methods performed) is found in the middle of the article (line 152-153). This must appear immediately before the figure, and not alone in the middle of the text. 

R: We appreciate your observation. The title of the figure was inserted in the text following the journal rules in the guide for authors. (Figure captions must be inserted in the text of the manuscript, immediately following the paragraph in which the figure is first cited (read order). Do not include captions as part of the figure files themselves or submit them in a separate document).

The authors return the results data in the discussion, once again, the indication of the tables. I believe that this indication of the tables in the discussion is not necessary (lines 234 and 239), given that it was made in the results.

R: We excluded the indication of the tables and the p-values from the discussion as suggested by the reviewer. Thanks for watching this mistake and we apologize for not fixing it in the first time.

This is an article relevant to the field of public health and the data here exposed are well presented. I take this opportunity to thank you for the invitation and opportunity to review this work. Nothing more to say.

Kind regards,

F. S.

Reviewer #2: (No Response)

---

## [Editor Report · Decision Letter 2]

24 Sep 2020

Prevalence and factors associated with Trichomonas vaginalis infection in indigenous Brazilian women

PONE-D-20-17319R2

Dear Dr. Marchioro,

We’re pleased to inform you that your manuscript has been judged scientifically suitable for publication and will be formally accepted for publication once it meets all outstanding technical requirements.

Kind regards,

Catherine E Oldenburg

Academic Editor

PLOS ONE
---

## [Editor Report · Acceptance letter]

7 Oct 2020

PONE-D-20-17319R2 

Prevalence and factors associated with *Trichomonas vaginalis* infection in indigenous Brazilian women

Dear Dr. Marchioro:

I'm pleased to inform you that your manuscript has been deemed suitable for publication in PLOS ONE. Congratulations! Your manuscript is now with our production department. 

Kind regards, 

on behalf of

Dr. Catherine E Oldenburg 

Academic Editor

PLOS ONE